# Apoptotic Cell-Derived CD14(+) Microparticles Promote the Phagocytic Activity of Neutrophilic Precursor Cells in the Phagocytosis of Apoptotic Cells

**DOI:** 10.3390/cells12151983

**Published:** 2023-08-01

**Authors:** Yu-Chieh Lin, Wen-Hui Tsai, Shao-Chi Chang, Hui-Chi Hsu

**Affiliations:** 1Department of Physiology, School of Medicine, National Yang-Ming Chiao-Tung University, Taipei 112, Taiwan; b117094033@gmail.com (Y.-C.L.);; 2Sleep Medicine Center, Division of Chest Medicine, Taichung Tzu Chi Hospital, Taichung 427, Taiwan; 3Department of Respiratory Therapy, Taipei Medical University, Taipei 106, Taiwan; tsaiwh@gmail.com; 4Department of Medicine, School of Medicine, National Yang-Ming Chiao-Tung University, Taipei 112, Taiwan; 5Division of General Medicine, Department of Medicine, Taipei Veterans General Hospital, Taipei 112, Taiwan; 6Division of Hematology & Oncology, Department of Medicine, Cheng-Hsin General Hospital, Taipei 112, Taiwan

**Keywords:** apoptotic cells, CD14, differentiation syndrome, granulocytic differentiation, microparticles, neutrophilic precursor cells, phagocytosis, resolution, retinoic acid

## Abstract

Membranous CD14 is crucial in the phagocytic activity of neutrophils. However, the role of CD14(+) microparticles (MPs) derived from apoptotic neutrophils (apo-MP) during the phagocytic process is not clear. All trans-retinoic acid (ATRA) induces acute promyelocytic leukemic NB4 cells along granulocytic differentiation. In this study, we investigated the role of CD14(+)apo-MP in the cell–cell interaction during the phagocytic process of apoptotic cells by viable ATRA-NB4 cells. We firstly demonstrate that CD14 expression and phagocytic activity of NB4 cells were upregulated simultaneously after ATRA treatment in a time-dependent manner, and both were significantly enhanced via concurrent lipopolysaccharide treatment. The phagocytic activity of ATRA-NB4 cells and lipopolysaccharide-treated ATRA-NB4 cells were both significantly attenuated by pre-treating cells with an antibody specific to either CD14 or TLR4. Further flow cytometric analysis demonstrates that apoptotic ATRA-NB4 cells release CD14(+)apo-MP in an idarubicin dosage-dependent manner. Both CD14 expression and the phagocytic activity of viable ATRA-NB4 cells were significantly enhanced after incubation with apo-MP harvested from apoptotic ATRA-NB4 cells, and the apo-MP-enhanced phagocytic activity was significantly attenuated by pre-treating apo-MP with an anti-CD14 antibody before incubation with viable cells. We conclude that CD14(+)apo-MP derived from apoptotic ATRA-NB4 cells promotes the phagocytic activity of viable ATRA-NB4 cells in engulfing apoptotic cells.

## 1. Introduction

In the early inflammation phase of acute lung injury, alveolar macrophages strategically located in the alveolar space destroy invading pathogens, including microorganisms and environmental toxins, while numerous neutrophils and monocytes are rapidly recruited from the bloodstream to the alveolar space [1]. Neutrophils are the primary effector cells destroying invading pathogens at the infection site [2]. Although the life span of neutrophils is extended in inflamed tissues, the apoptosis of numerous recruited neutrophils has been documented in the inflammatory sites [3]. Early recognition and efferocytic phagocytosis of dying neutrophils by alveolar macrophages are crucial in preventing the uncontrolled release of toxic substances from dead neutrophils to cause lung tissue destruction [4,5]. The efferocytic engulfment of dying cells by macrophages requires the formation of the phagocytic synapse between apoptotic cells and macrophages, and these are regulated by a network of “find-me”, “eat-me”, and “do not eat me” signals; bridging molecules; and specialized phagocytic receptors on the surface of both cells [6,7]. However, the number of alveolar macrophages and recruited monocytes are relatively less compared to that of recruited neutrophils at inflammatory sites of an acute lung injury. A recent study has reported that activated recruited neutrophils also play a crucial role in the clearance of apoptotic cells at the inflammatory sites [8]. In severe infections or inflammations, neutrophilic precursor cells (NPCs) are released into circulation, which eventually are recruited into inflammation sites. However, it is still not clear whether NPCs engage in the phagocytosis of either bacteria or apoptotic cells at the inflammatory sites.

Microparticles (MPs) are derived from membrane budding in neutrophils, macrophages, platelets, endothelial cells, and erythrocytes during cellular activation or during the later stages of apoptosis [9]. MPs play an important role in the cell–cell interactions among inflammatory cells [10]. The surface of MPs carries antigens and adhesion molecules from their original cell membrane, and these are able to mediate intercellular cross-talk by transferring receptors, antigens, and cytokines from donor to recipient cells [9]. Recently, we have reported that MPs play a crucial role in the cell–cell interaction in the phagocytosis of apoptotic cells via macrophages during the resolution phase of inflammation [11,12,13].

CD14 has been known as a macrophage-specific differentiation antigen, which is expressed mainly by macrophages and (to a lesser extent) by neutrophils [14]. In reacting to insults such as bacterial lipopolysaccharide (LPS), membranous CD14 and Toll-like receptor 4 (*TLR4*)/myeloid differentiation factor 2 (*MD-2*) complex on macrophages are brought in close proximity inside lipid rafts to (1) activate the TIRAP-MyD88-dependent pathway for the activation of NF-κB, leading to the enhancement of phagocytic activity and the release of pro-inflammatory cytokines such as TNF-α, IL-6, pro-IL18, and pro-IL-1β; (2) promote endocytosis of TLR4 and activate IRF3 signaling pathway, leading to enhanced type-I interferon expression [15,16]. On the other hand, CD14 on macrophages also plays a crucial role in the recognition and phagocytosis of apoptotic cells without inciting inflammation, and this is in stark contrast to its role in eliciting pro-inflammatory responses following binding with LPS [17]. However, the role of CD14 in the phagocytic activity of NPC has been rarely studied. Recently, we and others reported that CX3CL1 on microparticles (MPs) derived from apoptotic cells (apo-MP) acts as a “find-me” signal to attract macrophage transmigration toward apoptotic cells and to promote the phagocytic activity of macrophages, indicating the crucial role of apo-MP in the cell–cell interaction during the process of the efferocytic clearance of apoptotic cells [11,12,18]. However, it is still not clear whether CD14(+)apo-MP, derived from apoptotic NPC, plays any role in the phagocytosis of apoptotic cells.

All-trans retinoic acid (ATRA) has been the standard care in the treatment of acute promyelocytic leukemia (APL) by inducing APL cells along granulocytic differentiation into mature neutrophils. However, this treatment can be complicated by the occurrence of a differentiation syndrome (DS; also known as retinoic acid syndrome or cytokine storm) in up to 25% of treated patients, which generally occurs 7 to 12 days after starting inductive ATRA treatment in APL patients [19]. However, the risk factor for the development of DS and how to prevent the development of DS in APL patients is still not clear. Previous studies have indicated that ATRA-induced granulocytic differentiation in APL cells is associated with (a) increased production of pro-inflammatory cytokines, such as interleukin (IL)-1, IL-6, IL-8, monocyte chemotactic protein-1, tumor necrosis factor-α, and others [20,21,22,23]; (b) an excess release of cathepsin G to enhance capillary permeability and damage [24]; and (c) the upregulation of leukocyte integrins to promote their adherence to capillary endothelium and organ infiltration [25]. These contribute to the massive transmigration of ATRA-treated APL (ATRA-APL) cells into alveolar space, resulting in the development of diffuse pulmonary infiltrate, pleural, or pericardial effusion in patients with DS [19,26]. Once entering the resolution phase of DS, numerous apoptotic ATRA-APL cells are needed to be cleared from alveolar spaces to restore the normal gas exchange function. We reported recently that both viable ATRA-APL cells and alveolar macrophages play an important role in the phagocytic clearance of apoptotic ATRA-APL cells [11]. Therefore, we hypothesize that apoptotic ATRA-APL cells release CD14(+)apo-MP to orchestrate the phagocytic clearance of apoptotic cells by viable ATRA-APL cells during the resolution phase in APL patients with DS. In this study, we used ATRA-APL NB4 (ATRA-NB4) cells as a study model to investigate (a) the role of membranous CD14 on ATRA-NB4 cells in the development of phagocytic activity and (b) the role of CD14(+) apo-MP derived from apoptotic ATRA-NB4 cells in the phagocytic clearance of apoptotic cells by viable ATRA-NB4 cells.

## 2. Materials and Methods

### 2.1. Cell Culture and the Preparation of Conditioned Medium (CM)

NB4 cells are a human APL cell line, which was cultured in RPMI-1640 medium (GIBCO, Grand Island, NY, USA) as described previously [27]. NB4 cells (1 × 10^5^ cells/mL) were treated with ATRA (1 µM; Sigma, St. Louis, MO, USA) with or without LPS (100 ng/mL) for 6 days. ATRA-NB4 cells were adjusted to the cell densities of 1 × 10^6^ cells/mL before further experiments.

To prepare CM, supernatants from ATRA-NB4 cell cultures were harvested and then centrifuged at 250× *g* for 5 min to remove all cellular components. Finally, these CMs either underwent MP purification or were stored as aliquots at −20 °C.

### 2.2. Preparation of Apoptotic Cells

To induce apoptosis, ATRA-NB4 cells were treated with idarubicin (Ida; Sigma, St. Louis, MO, USA) at 5–50 nM/mL for 4 h and incubated at 37 °C [28]. Thereafter, the washed, Ida-treated ATRA-NB4 (Ida-ATRA-NB4) cells were incubated with both annexin V (NXPE; R&D Systems, Minneapolis, MN, USA) and 7-amino-Actinomycin-D (7-AAD; BD Bioscience Pharmingen, San Diego, CA, USA) for double staining before analysis via flow cytometry to measure the percentage of early apoptotic cells and late apoptotic cells, respectively [29].

### 2.3. MP Preparation and Flow Cytometric Analysis

We harvested MP from ATRA-NB4 cell cultures, as reported by Gasser et al. and Tsai et al. [30,31]. The MP pellet was washed once and re-suspended in PBS. For flow cytometric analysis, the MPs were stained with annexin V (NXPE; R&D Systems, Minneapolis, MN, USA) or anti-CD14 antibodies (R&D System, Minneapolis, MN, USA). The flow cytometric analysis of the MP preparations showed the expected heterogeneous populations, with sizes varying approximately between 0.1 and 2 µm as verified using control latex beads (2 µm each; ACBP-20-10; Becton Dickson; San Jose, CA, USA). The numbers of MPs within the same gated area of MPs in the flow cytometric dot plots were then calculated.

### 2.4. Flow Cytometric Analysis of ATRA-NB4 Cells

The non-permeabilized ATRA-NB4 cells were stained with either CD14 monoclonal antibody (R&D System, Minneapolis, MN, USA) or TLR4 monoclonal antibody (R&D System, Minneapolis, MN, USA), respectively, before analysis using FAC Scan as reported previously [13].

### 2.5. Assess the Phagocytic Engulfment of Apoptotic Cells by Flow Cytometric Analysis

ATRA-NB4 cells were first treated with one of the following: LPS, Anti-CD14 (R&D, Minneapolis, MN, USA), Anti-TLR4 (R&D, Minneapolis, MN, USA), Cytochalasin D (Sigma, St. Louis, MO, USA), Bay11–7082 (InvivoGen, San Diego, CA, USA), or SP600125 (Invivogen, San Diego, CA, USA). Thereafter, ATRA-NB4 cells were washed with 5% PBS solution twice. Subsequently, the ATRA-NB4 cells were incubated with either fluorescein isothiocyanate (FITC) latex beads (Polysciences, Inc., Warrington, PA, USA, average diameter 2 m) at a ratio of 1:10 (cells: latex beads) for 1 h or PKH−26 (Sigma-Aldrich, St. Louis, MO, USA) labeled Ida-ATRA-NB4 cells (2 × 10^6^) for 30 min before determining the amount of phagocytosis using a BD FACScan [13,32]. The results were expressed as (a) the percentage (%) of ATRA-NB4 cells with phagocytic activity in engulfing either latex beads or apoptotic cells and (b) a phagocytosis index that indicates a fold increase relative to the phagocytic activity of untreated ATRA-NB4 cells. Part of ATRA-NB4 cells, pre-incubated with either latex beads or labeled apoptotic cells, were examined under confocal microscopy (Fluoview; Olympus, Tokyo, Japan).

### 2.6. Statistical Analysis

The results were evaluated via the Kolmogorov–Smirnov test and/or the Shapiro–Wilk test for normal distribution. Thereafter, the results were evaluated via one-way ANOVA followed by either the Fisher’s least significant difference (LSD) procedure where appropriate or the Dunnett post hoc test. A value of *p* < 0.05 was considered significant. All data were analyzed using SPSS software version 23 (IBM SPSS Statistics, Armonk, NY, USA). All results are presented as mean ± SD.

## 3. Results

### 3.1. ATRA Induces the Phagocytic Activity of NB4 Cells in Engulfing Latex Beads in a CD14-Dependent Manner

We first determined the expression of CD14 and TLR4 on the surface of NB4 cells via flow cytometry. Figure 1A demonstrates that CD14 was minimally expressed on the surface of ATRA-untreated NB4 cells, and its level was progressively enhanced in a time-dependent manner when NB4 cells were treated with ATRA for 6 days (*p <* 0.001). However, TLR4 was constitutively expressed on the surface of ATRA-untreated NB4 cells, and its level was enhanced progressively in a time-dependent manner when NB4 cells were treated with ATRA for 6 days (*p <* 0.001; Figure 1B). Similarly, the percentage of ATRA-NB4 cells expressing both CD14 and TLR4 was also significantly increased in a time-dependent manner when NB4 cells were treated with ATRA for 6 days (*p <* 0.001; Figure 1C).

In determining the phagocytic activity of ATRA-NB4 cells, we first demonstrate that ATRA-NB4 cells engulfed the fluorescence-labeled latex beads via the fluorescence microscopic examination, as shown in Figure 2A,C. Thereafter, we determine their phagocytic activity via flow cytometry. Figure 2D shows that the phagocytic activity of ATRA-untreated NB4 cells was minimal, and its level was progressively enhanced in a time-dependent manner when NB4 cells were treated with ATRA for 6 days (*p <* 0.005). The enhanced phagocytic activity of ATRA-NB4 cells was significantly attenuated when ATRA-NB4 cells were pre-treated with a phagocytic inhibitor, cytochalasin D, before incubating latex beads for phagocytic assay. Next, we determined the phagocytic activity in those CD14(+) ATRA-NB4 cells via flow cytometry. Figure 2E demonstrates that, when NB4 cells were treated with ATRA for 6 days, the percentage of CD14(+) ATRA-NB4 cells associated with phagocytic activity increased progressively in a time-dependent manner (*p <* 0.001), and up to 74% of CD14(+) ATRA-NB4 cells had phagocytic activity in engulfing latex beads. Figure 3A further demonstrates that, when compared with the level of those cells without pre-treatment with an antibody, the phagocytic activity of ATRA-NB4 cells in engulfing latex beads was significantly attenuated when cells were pre-treated with either an anti-CD14 antibody or an anti-TLR4 antibody before incubation with labeled latex beads for phagocytic assay (*p <* 0.001 and *p <* 0.001; respectively), while no additional attenuation of phagocytic activity was found in those cells pre-treated with both anti-CD14 antibody and anti-TLR4 antibody. Collectively, ATRA induces the expression of CD14 on the surface of NB4 cells while the latter cells are also induced into the process of granulocytic differentiation [33], and CD14 expression contributes to the development of phagocytic activity in ATRA-NB4 cells. Thereafter, we determined the signal transduction pathway underlying the phagocytic activity of ATRA-NB4 cells. Figure 3B demonstrates that, compared with those cells without any inhibitor pre-treatment, the phagocytic activity of ATRA-NB4 cells was significantly attenuated via pre-treatment with either a NF-κB inhibitor (BAY11-7082; *p <* 0.001) or an IRF3 inhibitor (SP; *p <* 0.001) before incubation with fluorescence-labeled latex beads for phagocytic analysis, and this level was further attenuated by pre-treating those cells with both BAY11-7082 and SP when compared with the levels of those cells pre-treated with either BAY11-9082 alone or SP alone (*p <* 0.001 and *p <* 0.001; respectively).

### 3.2. LPS Enhances the Phagocytic Activity of ATRA-NB4 Cells in Engulfing Latex Beads in a CD14-Dependent Manner

Figure 3C demonstrates that, compared with LPS-untreated ATRA-NB4 cells, LPS treatment significantly increased the percentage of CD14(+) ATRA-NB4 cells (*p <* 0.01) while significantly decreasing the percentage of TLR4(+) cells (*p <* 0.001) but had no significant change in the percentage of cells expressing both CD14 and TLR4.

We next determined the effect of LPS on the phagocytic activity of ATRA-NB4 cells in engulfing latex beads. Figure 3A demonstrates that the phagocytic activity of ATRA-NB4 cells in engulfing latex beads was significantly enhanced with LPS treatment compared with that of LPS-untreated cells (*p <* 0.001), and this level was significantly attenuated when the LPS-treated cells were pre-treated with an anti-CD14 antibody alone, anti-TLR4 antibody alone, or combination of both antibodies before incubation with latex beads for phagocytic assay (*p <* 0.001, *p <* 0.001, and *p <* 0.001, respectively), and the level of cells pre-treated with both anti-CD14 and anti-TLR4 antibodies was not significantly different from the level of those cells pre-treated with either anti-CD14 antibody alone and anti-TLR4 antibody alone. Collectively, our results indicate that LPS enhanced the phagocytic activity of ATRA-NB4 cells in a CD14-dependent and/or TLR4-dependent manner. Figure 3B further demonstrates that, as compared with those cells without any inhibitor pre-treatment, the phagocytic activity of LPS-treated ATRA-NB4 cells was significantly attenuated via pre-treatment with either BAY11-7082 alone (*p <* 0.001) or SP alone (*p <* 0.001) before incubation with latex beads for phagocytic analysis, and the level of cells pre-treated with both BAY11-7082 and SP was significantly attenuated when compared with the level of those cells pre-treated with either BAY11-9082 alone or SP alone (*p <* 0.01 and *p <* 0.001, respectively).

### 3.3. CD14 Mediates the Phagocytic Activity of ATRA-NB4 Cells in Engulfing Apoptotic Cells

We next determined the effect of idarubicin on the induction of apoptosis in ATRA-NB4 cells via flow cytometry. After 4 h of idarubicin treatment in ATRA-NB4 cells, the percentage of cells in the early apoptotic stage increased in a dosage-dependent manner (*p <* 0.001), while the levels of those cells in the late apoptotic stage showed no significant difference among different dosages of idarubicin (Table 1). Thereafter, we incubated viable ATRA-NB4 cells with washed Ida-treated ATRA-NB4 (Ida-ATRA-NB4) cells to determine their phagocytic activity in engulfing apoptotic cells via flow cytometric analysis. Our data demonstrate that, after treatment with ATRA for 6 days, the percentage of ATRA-NB4 cells with phagocytic engulfment of apoptotic cells (mean: 11.5%; range: 9.6–13.1%) is significantly higher than that in those cells pre-treated with cytochalasin D (mean; 6.4%; range: 5.4–8.2%) (*p <* 0.05). In addition, Figure 4A demonstrates that, when NB4 cells were treated with ATRA for 6 days, the phagocytic activity of ATRA-NB4 cells was significantly enhanced in a time-dependent manner (*p <* 0.001), and the enhanced level was significantly attenuated when ATRA-NB4 cells were pre-treated with cytochalasin D before incubation with apoptotic cells for phagocytic assay (*p <* 0.05). Next, we incubated ATRA-NB4 cells with apoptotic cells that had been induced into the process of apoptosis with different dosages of idarubicin. Our data demonstrate that, via incubation with those apoptotic cells induced by 50 nM idarubicin, the percentage of viable ATRA-NB4 cells with the phagocytic engulfment of apoptotic cells is significantly higher than the level of ATRA-NB4 cells pre-treated with cytochalasin D [(mean: 11.8%; range: 9.9–15.4%) vs. (mean: 5.3%; range: 4.3%–6.8%), respectively] (*p <* 0.05). Furthermore, Figure 4B demonstrates that the phagocytic activity of ATRA-NB4 cells in engulfing Ida-ATRA-NB4 cells was significantly enhanced in an idarubicin dosage-dependent manner when ATRA-NB4 cells were induced into apoptotic process with a higher dosage of idarubicin (*p <* 0.01), and this enhanced phagocytic activity was also significantly attenuated when those viable ATRA-NB4 cells were pre-treated with cytochalasin D before incubation with apoptotic cells for phagocytic assay (*p <* 0.05). Thereafter, Figure 4C demonstrates that, compared with cells without antibody treatment, the phagocytic activity of ATRA-NB4 cells was significantly attenuated via pre-treatment with an anti-CD14 antibody alone, anti-TLR4 antibody alone, or both anti-CD14 antibody and anti-TLR4 antibodies before incubation with the apoptotic cells for phagocytic assay (*p <* 0.001, *p <* 0.001, and *p <* 0.001, respectively). The levels of phagocytic activity of ATRA-NB4 cells pre-treated with both anti-CD14 antibody and anti-TLR4 have no significant differences when compared with the level of those cells pre-treated with anti-CD14 alone. However, this level was significantly attenuated when compared with the levels of those cells pre-treated with anti-TLR4 antibody alone (*p <* 0.01). In addition, Figure 4D demonstrates that, as compared with cells without treatment with an inhibitor, the phagocytic activity of ATRA-NB4 cells was significantly attenuated via pre-treatment with BAY11-7082 before incubation with apoptotic cells for phagocytic assay (*p <* 0.005), while the levels in those cells pre-treated with SP alone showed no significant differences. Furthermore, the level of phagocytic activity of ATRA-NB4 cells pre-treated with both BAY11-7082 and SP was significantly attenuated when compared with that of those cells pre-treated with SP alone (*p <* 0.001) but had no significant difference when compared with the level of those cells pre-treated with BAY11-7082 alone.

### 3.4. Apoptotic Cells-Derived CD14(+)apo-MP Enhances the Phagocytic Activity of Viable ATRA-NB4 Cells in Engulfing Apoptotic Cells

Finally, we harvested apo-MP from the conditioning medium of Ida-ATRA-NB4 cell culture to determine their role in the phagocytic activity of viable ATRA-NB4 cells in engulfing apoptotic cells via flow cytometric analysis. Figure 5A demonstrates that the level of apo-MP and CD14(+)apo-MP released by Ida-ATRA-NB4 cells increased progressively in an Ida-dosage dependent manner (*p <* 0.001 and *p <* 0.01, respectively) when ATRA-NB4 cells were induced into apoptosis via pre-treatment with an increased dosage of idarubicin. Thereafter, viable ATRA-NB4 cells were incubated with florescence labeled CD14(+)apo-MP [CD14(+)apo-MP*] to determine their surface expression of CD14 and phagocytic activity. As shown in Figure 5B,C, the surface expression of CD14 on viable ATRA-NB4 cells was significantly enhanced in an idarubicin dosage-dependent manner when viable ATRA-NB4 cells were incubated with CD14(+)apo-MP* harvested from those cells treated with a higher dosage of idarubicin (*p <* 0.001). In addition, the phagocytic activity of ATRA-NB4 cells pre-incubated with apo-MP was significantly enhanced compared with the levels of those cells pre-incubated with either vehicle or MP harvested from Ida-untreated ATRA-NB4 cells (*p <* 0.001 and *p <* 0.001, respectively, Figure 5D), and the apo-MP-enhanced phagocytic activity was significantly attenuated when that apo-MP was pre-treated with anti-CD14 antibody before incubation with viable ATRA-NB4 cells for phagocytic assay (*p <* 0.001). Taken together, membranous CD14 of apo-MP derived from apoptotic ATRA-NB4 cells enhance the phagocytic activity of the viable ATRA-NB4 cells in engulfing apoptotic cells.

## 4. Discussion

In this study, we provide the novel finding that (1) membranous CD14 of ATRA-NB4 cells contribute to the development of phagocytic activity during the granulocytic differentiation process and (2) CD14(+) apo-MP derived from apoptotic ATRA-NB4 cells enhance the phagocytic activity of recipient viable ATRA-NB4 cells in a CD14-dependent manner. To support our annotation, we demonstrate that (1) both membranous CD14 expression and phagocytic activity were upregulated simultaneously in NB4 cells after treatment with ATRA in a time-dependent manner (Figure 1 and Figure 2), while those cells develop other functions of mature granulocytes such as cytokine release, transmigration, and adhesion activities [23]; and (2) the phagocytic activity of ATRA-NB4 cells was attenuated via pre-treatment with an anti-CD14 antibody before phagocytic assay (Figure 3A). Consistent with our results, CD14 also contributes to the development of phagocytic activity during the monocytic differentiation process in both U937 and HL-60 cells after induction by dimethyl sulfoxide and active 1,25-dihydroxyvitamin D3, respectively [34]. CD14 expression in neutrophils is 10 times lesser than in macrophages and monocytes [14]; this partially explains that CD14 expression was only detectable in around 15 percent of NB4 cells treated with ATRA for 6 days (Figure 1C), though a longer duration of ATRA treatment is needed for NB4 cells to differentiate into mature granulocytes [35]. In parallel with this, Kim et al. reported recently that membranous CD14 of phagocytes recognize phosphatidylinositides and externalize on apoptotic cells as eat-me signals [36], while CD14 also acts as a rapid-acting tethering molecule by interacting with intercellular adhesion molecule-3 on the apoptotic cells to facilitate the recognition and tethering of apoptotic cells [37], and these contribute to the formation of phagocytic synapse between phosphatidylserine and ανβ3 integrins on apoptotic cells and phagocytes, respectively, for subsequent engulfing phagocytosis [38]. To further confirm the crucial role of CD14 in the clearance of apoptotic cells, Gregory et al. reported that apoptotic cells have been shown to accumulate in the tissues of CD14^−/−^ mice [39]. In addition, Thomas et al. reported that CD14 on macrophages relies on residue 11 for apoptotic cell tethering, whilst other residues are reported as key for LPS binding [40]. Regarding this, Figure 3 demonstrates that LPS enhances the phagocytic activity of ATRA-NB4 cells in engulfing latex beads in both CD14- and TLR4-dependent manner. In parallel with this, previous studies have reported that LPS induces de novo synthesis of CD14 and promotes the translocation of CD14 from the cytoplasmic reservoir to the surface membrane of mature neutrophils during bacterial infection [41,42,43]. In addition, our results demonstrate that the surface expression of TLR4 was down-regulated in the ATRA-NB4 cells after treatment with LPS (Figure 3C). This is in agreement with a previous study that the binding of LPS with membranous CD14 of macrophages promotes the endocytosis of TLR4 to redirect plasma membranous TLR4 to the endosomal compartment with the consequent activation of the TRAM-TRIF pathway and IRF3 signaling pathway, leading to type-I IFN production [15,16,44,45,46,47]. Taken together, our data imply that CD14 contributes to the phagocytic activity of ATRA-NB4 cells in the phagocytosis of bacteria or apoptotic cells at the inflammatory sites.

Our results further indicate that membranous CD14 of ATRA-NB4 cells mediates the phagocytosis of latex beads in a TLR4-dependent manner leading to the activation of both NF-κB and IRF-3 signaling pathway while mediating the phagocytosis of apoptotic cells in a TLR4-independent manner leading to the activation of the NF-κB signaling pathway, but not the IRF-3 pathway. To support our proposal, we demonstrate that the phagocytic activity of ATRA-NB4 cells in engulfing latex beads was significantly attenuated via the pre-treatment of ATRA-NB4 cells with either a NF-κB inhibitor or an IRF-3 inhibitor, while the phagocytic activity in engulfing apoptotic cells was only significantly attenuated via pre-treatment with a NF-κB inhibitor, but not with an IRF-3 inhibitor (Figure 3B and Figure 4D). This is in agreement with previous studies that CD14 on macrophages mediates effective phagocytosis of apoptotic cells in the absence of detectable TLR4, whilst binding and responsiveness to LPS require TLR4 [40]. In addition, Shiratsuchi et al. reported that TLR4^-/-^ macrophages take up apoptotic cells more efficiently than TLR4^+/+^ macrophages, and TLR-4 expression negatively correlates with CD14′s ability to tether apoptotic cells, possibly through competition for CD14 binding [40,48].

Thereafter, we address the crucial role of CD14(+)apo-MP in the phagocytosis of apoptotic cells by viable ATRA-NB4 cells. We have reported recently that apoptotic ATRA-NB4 cells release CX3CL1(+)apo-MP to (a) induce both viable ATRA-NB4 cells and alveolar macrophages transmigration toward apoptotic cells and (b) upregulate the MFG-E8 expression on alveolar macrophages as a bridge molecule to promote the binding between phosphatidylserine and phagocytic receptors (integrin αvβ3/5) on apoptotic cells and macrophages, respectively, for subsequent phagocytosis [11,12,13]. In this study, we provide the novel finding that apoptotic ATRA-NB4 cells release CD14(+) apo-MP to promote the recipient viable ATRA-NB4 cells in the early recognition and tethering of apoptotic cells for subsequent phagocytosis. Regarding this, we demonstrate that (a) CD14(+) apo-MP was released by apoptotic ATRA-NB4 cells in an idarubicin-dosage dependent manner, (b) membranous CD14 expression of viable ATRA-NB4 cells was enhanced after incubation with apo-MP harvested from apoptotic ATRA-NB4 cells, and (c) the apo-MP-enhanced phagocytic activity of viable ATRA-NB4 cells was significantly attenuated when that apo-MP was pre-treated with an anti-CD14 antibody before incubation with viable cells for phagocytic assay (Figure 5D). Our data imply that membranous CD14 on apo-MP, derived from apoptotic ATRA-NB4 cells, was transferred to the membrane of recipient viable ATRA-NB4 cells via adhesion and/or a fusing mechanism [12]. Besides CD14, apoptotic ATRA-NB4 cell-derived apo-MP also enhance MFG-E8 expression on recipient viable ATRA-NB4 cells via CX3CL1-CX3CR1 axis to promote the binding between MFG-E8 and integrin ανβ3/5, resulting in the induction of STAT3-mediated SOCS3 activation to inhibit both NF-κB signaling pathway and pro-inflammatory cytokine production, and this contributes to the phagocytosis of apoptotic cells without inducing inflammation [49]. Besides membranous CD14 on phagocytes and their MP, there is another soluble form of CD14 (sCD14), which appears after the shedding of membranous CD14 or is directly secreted from intracellular vesicles [50]. The sCD14 has differing effects on LPS-related inflammatory signaling [15]. On one hand, sCD14 can enhance LPS-related inflammation in cells both harboring and lacking membrane-bound CD14 [51,52]. On the other hand, high levels of sCD14 can buffer LPS by promoting its transfer to lipoprotein particles and preventing LPS binding with membranous CD14 of monocytes/macrophages [53]. During acute inflammation, activated neutrophils also release human neutrophil elastase to promote the release of sCD14 by cleaving membranous CD14 of macrophage, resulting in reducing the phagocytic recognition of apoptotic cells but not blocking the engulfment of bacteria or latex beads [54]. Further investigation is warranted to study the interaction between sCD14 and CD14(+)apo-MP during the inflammatory process.

In the clinical setting, we speculate that occult or sub-clinical bacterial infection in the alveolar spaces in APL patients may provoke ATRA-APL cells to transmigrate from the blood stream into alveolar spaces via chemokine gradients such as IL-8 and MCP [19,20]. Once arriving at alveolar space, membranous CD14 of ATRA-APL cells contributes to the phagocytic clearance of bacteria by binding with LPS on bacteria, and this also provokes the release of pro-inflammatory cytokines via the NF-κB signal transduction pathway to recruit more ATRA-APL cells from the blood stream into alveolar spaces for the development of DS in APL patients under induction ATRA treatment [16,20,22]. Therefore, we hypothesize that occult bacterial infection of the lung in ATRA-untreated APL patients may be a potential risk factor for the development of DS when they receive ATRA induction treatment. This warrants further clinical trials to study whether prophylactic antibiotics treatment is able to prevent the development of DS in APL patients under inductive ATRA treatment. On the other hand, CD14 on ATRA-APL cells also plays a crucial role during the resolution phase of DS in APL patients because the pool of normal monocytes/macrophages recruited into the alveolar spaces is markedly suppressed by malignant APL cell clones in the bone marrow. Our data imply that apoptotic ATRA-APL cells-derived CD14(+)apo-MP promote the recipient viable ATRA-APL cells in the efferocytic engulfment of apoptotic cells, and this prevents the release of pro-inflammatory cytokines from apoptotic cells and the recruitment of ATRA-APL cells into alveolar space during the resolution phase of DS in APL patients. This warrants further clinical study to investigate the role of CD14(+)MP in the bronchoalveolar lavage in APL patients with DS. Finally, our data also confirm the important role of CD14 on the phagocytic activity of those NPC when they are recruited into the inflammatory site in patients with an overwhelming bacterial infection. 

## Figures and Tables

**Figure 1 cells-12-01983-f001:**
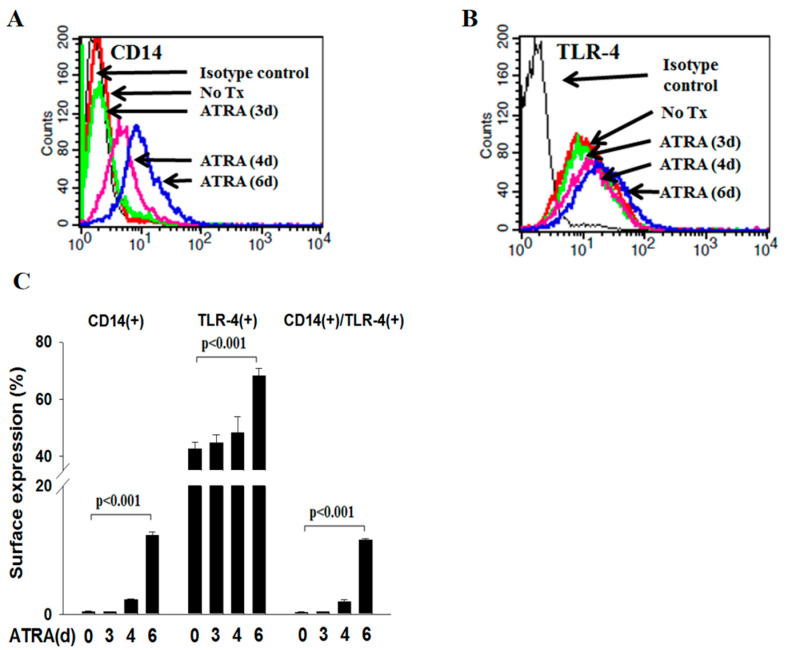
ATRA induced the surface expression of CD14 and TLR4 on NB4 cells. NB4 cells were treated with ATRA for 6 days before determining the surface expression of CD14 and TLR4 via flow cytometric analysis. (**A**,**B**) This is a representative picture from four independent experiments. (**C**) This presents the percentage of ATRA-NB4 cells expressing CD14 only, TLR4 only, or both CD14 and TLR4, which represents the mean ± SD from 4 independent experiments.

**Figure 2 cells-12-01983-f002:**
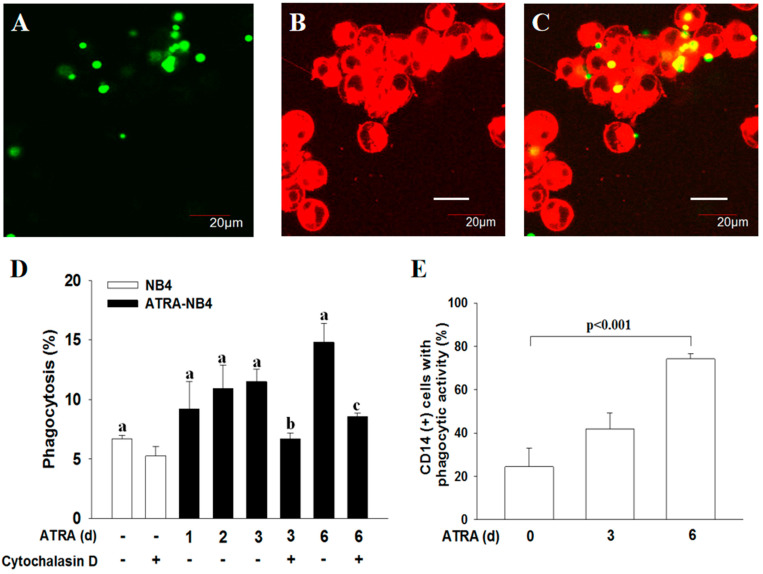
CD14 contributes to the development of phagocytic activity of ATRA-NB4 cells in engulfing latex beads. (**A**–**C**) Phagocytosis determined using confocal microscope. NB4 cells were treated with ATRA for 6 days before incubation with FITC-labeled latex beads for phagocytosis assay. The FITC-labeled latex beads (green: **A**) and PKH26-labeled ATRA-NB4 cells (red; **B**) were co-cultured for 2 h before being photographed using fluorescent microscope. (**C**) Overlay of photographs A and B. Magnification × 300. This is the representative picture of 4 independent experiments. (**D**,**E**) Phagocytic activity of ATRA-NB4 cells was determined via flow cytometric analysis. (**D**) NB4 cells (blank bar) were first treated with ATRA for 6 days (ATRA-NB4; black bar) before incubation with fluorescence-labeled latex beads in the presence or absence of cytochalasin D to determine their phagocytic activity. The results are expressed as the percentage of cells with phagocytic activity, which are the means ± SD from four independent experiments. a: *p <* 0.005; b: *p <* 0.05 vs. ATRA-NB4 cells (3 days) without cytochalasin D treatment; c: *p <* 0.01 vs. ATRA-NB4 cells (6 days) without cytochalasin D treatment. (**E**) NB4 cells were first treated with ATRA for 6 days before incubation with fluorescence-labeled latex beads to determine their CD14 expression and phagocytic activity. The results are expressed as the percentage of CD14(+) cells with phagocytic activity, which are the means ± SD from seven independent experiments.

**Figure 3 cells-12-01983-f003:**
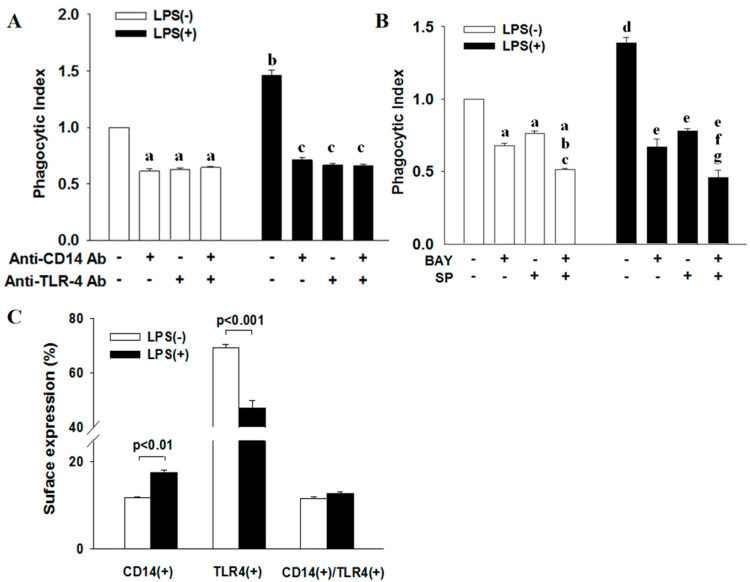
CD14 and TLR4 work cooperatively in the phagocytic activity of ATRA-NB4 cells in engulfing latex beads. (**A**,**B**) NB4 cells were firstly treated with either ATRA (blank bar) or ATRA + LPS (black bar) for 6 days before incubation with fluorescence-labeled latex beads, in the presence of (**A**) anti-CD14 antibody and/or anti-TLR4 antibody or (**B**) NF-κB inhibitor (BAY11-7082) and/or IRF3 inhibitor (SP), to determine their phagocytic activity via flow cytometric analysis. The results of phagocytic activity were expressed as phagocytosis index indicating a fold increase relative to the phagocytic activity of ATRA-NB4 cells without any treatment (Index bar = Blank left bar 1; **A**,**B**). (**A**) a: *p <* 0.001 vs. index; b: *p <* 0.001 vs. index; c: *p <* 0.001 vs. cells + LPS only (Black left bar 1). (**B**) a: *p <* 0.001 vs. index; b: *p <* 0.001 vs. cells + BAY only; c: *p <* 0.001 vs. cells + SP only; d: *p <* 0.001 vs. index; e: *p <* 0.001 vs. cells + LPS only (Black left bar 1); f: *p <* 0.01 vs. cells + LPS + BAY; g: *p <* 0.001 vs. cells + LPS + SP. (**C**) NB4 cells were treated with either ATRA only (blank bar) or ATRA + LPS (black bar) before determining the surface expression of CD14 and TLR4 via flow cytometric analysis. The results are the means ± SD from four (**A**,**B**) and seven (**C**) independent experiments, respectively.

**Figure 4 cells-12-01983-f004:**
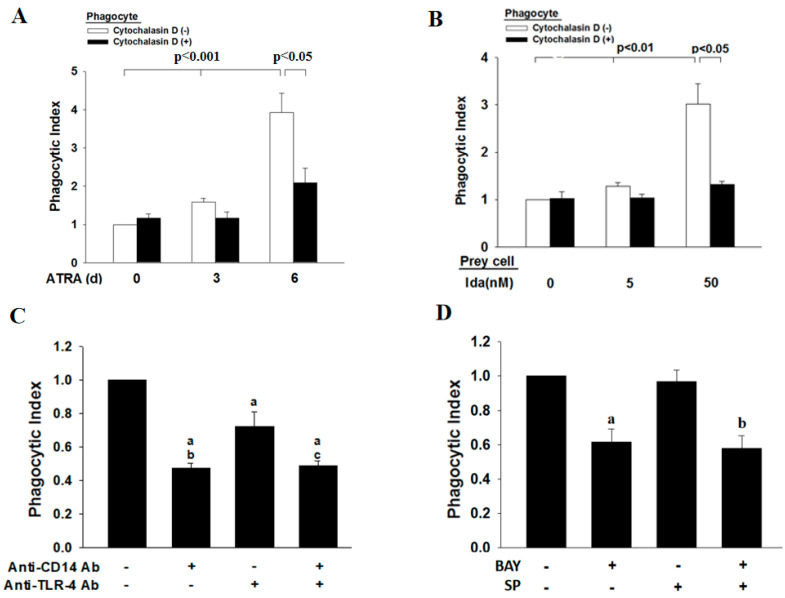
CD14 mediates the phagocytic activity of ATRA-NB4 cells in engulfing apoptotic cells. NB4 cells were treated with ATRA for 6 days before incubation with apoptotic cells for phagocytosis assay. (**A**,**B**) ATRA-NB4 cells, in the absence (Blank) or presence (Black) of cytochalasin D treatment, were incubated with apoptotic cells induced by either (**A**) a fixed dosage or (**B**) a different dosage of idarubicin before determination of phagocytic activity. (**C**,**D**) ATRA-NB4 cells were pre-treated with (**C**) an anti-CD14 antibody and/or anti-TLR4 antibody or (**D**) an NF-κB inhibitor (BAY) or an IRF3 inhibitor (SP) before incubation with apoptotic cells to determine their phagocytic activity. The results are expressed as phagocytic index indicating a fold increase relative to phagocytic activity of ATRA-untreated NB4 cells as control (**A**; left blank bar 1), cytochalasin D-untreated ATRA-NB4 cells as control (**B**; left blank bar) and ATRA-NB4 cells without any treatment (**C**,**D**: Index bar: left 1^st^ bar). (**C**) a: *p <* 0.001 vs. index bar; b,c: *p <* 0.01 vs. cells (+anti-TLR4 only). (**D**) a: *p <* 0.005 vs. index bar; b: *p <* 0.001 vs. index bar. The results are the means ± SD from four (**A**,**B**,**D**) and six (**C**) independent experiments.

**Figure 5 cells-12-01983-f005:**
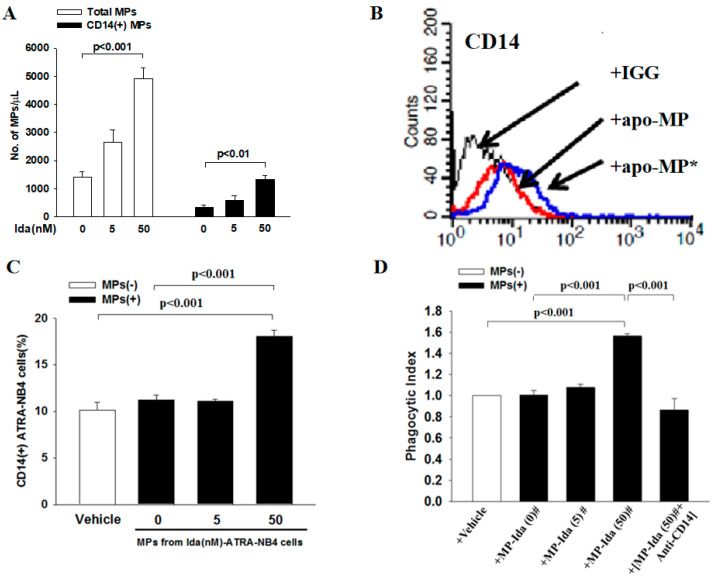
Apoptotic cells-derived CD14(+)MPs enhanced the phagocytic activity of ATRA-NB4 cells in engulfing apoptotic cells. Apo-MPs were harvested from 20 mL conditioning medium of ATRA-NB4 cells pre-treated with a different dosage of idarubicin. (**A**) The total numbers of apo-MPs (Blank bar) and CD14(+) apo-MPs (Black bar) were calculated using flow cytometric analysis. (**B**) Apo-MP harvested from Ida-ATRA-NB4 cells were either labeled (apo-MP*) or not labeled (apo-MP) with fluorescent FITC-conjugated CD14 monoclonal antibodies before incubation with viable ATRA-NB4 cells for subsequent determination of CD14 surface expression via flow cytometry analysis. This is a representative picture from four independent experiments. (**C**,**D**) ATRA-NB4 cells were firstly incubated with either vehicle or apo-MP harvested from a conditioning medium of ATRA-NB4 cells pre-treated with a different dosage of idarubicin [MP-Ida(dosage)#] before subsequent incubation with apoptotic cells to determine their phagocytosis activity via flow cytometry analysis. Part of apo-MP was pre-treated with CD14 monoclonal antibody [MP-Ida(50)# + Anti-CD14] before incubation with ATRA-NB4 cells. The results are the means ± SD from four independent experiments (**A**,**C**,**D**).

**Table 1 cells-12-01983-t001:** Idarubicin induces ATRA-NB4 cells into the process of apoptosis.

Idarubicin (nM)	Early Apoptosis (%) *	Late Apoptosis (%) *
0	10.4 + 1.0	6.4 + 0.7
5	17.5 + 0.2	7.6 + 0.2
50	34.5 + 2.9	9.4 + 1.0
*p* value **	*p* < 0.001	*p* = 0.062

* Early apoptosis is defined as cells expressing Annexin V but not 7-AAD; late apoptosis is defined as cells expressing both annexin V and 7-AAD. ** ANOVA.

## Data Availability

Not applicable.

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
