# Peer review of "Apoptotic Cell-Derived CD14(+) Microparticles Promote the Phagocytic Activity of Neutrophilic Precursor Cells in the Phagocytosis of Apoptotic Cells"

_cells, 2023, doi:10.3390/cells12151983_

Round 1
Reviewer 1 Report
The manuscript “Apoptotic cells-derived CD14(+) microparticles promote the phagocytic activity of neutrophilic precursor cells in the phagocytosis of apoptotic cells”, is a complete and comprehensive manuscript on the importance of the role of CD14 and CD14-carrying microparticles in the phagocytosis of apoptotic cells. Some minor reflections that should be worked on by the authors are:
- Throughout the manuscript some spaces between words are missing or, on the contrary, some that do not correspond have been introduced (indicated in color in the attached file).
-Some sentences are too long to be easily understandable.
- Figure legends are confused with the rest of the text.
Major observations
-Figure 1D. I did not find the description of this result in the text .
-I can see in the colocalization images (Figure 2C) some latex particles that have both colors, an orange part (as expected for colocalization) and a green part (no colocalization). Why does this happen? Is it really colocalization?
- In figure 2, why not use the phagocytosis inhibitor for the analysis of CD14 positive cells?
- The quality of figure 4 (A-C) must be increased because the colocalization is not well distinguished

There are minor fixes noted in the attached file
Author Response
- Throughout the manuscript some spaces between words are missing or, on the contrary, some that do not correspond have been introduced (indicated in color in the attached file).
-Some sentences are too long to be easily understandable.
- Figure legends are confused with the rest of the text.
Ans: We are very appreciated of the reviewer’s mention in the typing errors in our English, and we have corrected them accordingly. In addition, we have re-checked our figure legends and their connection with the text in the revised manuscript.
Major observations
-Figure 1D. I did not find the description of this result in the text .
Answer: I have moved Figure 1D to Figure 3C which has been described in L294 in the revised manuscript.
-I can see in the colocalization images (Figure 2C) some latex particles that have both colors, an orange part (as expected for colocalization) and a green part (no colocalization). Why does this happen? Is it really colocalization?
Ans: The orange latex beads were engulfed by ATRA-NB4 cells, while the green latex beads were not engulfed by cells.
- In figure 2, why not use the phagocytosis inhibitor for the analysis of CD14 positive cells?
Ans: The reviewer’s suggestion is great in the study design. However, sorting of CD14(+)ATRA-NB4 cells is not available in our lab due to limitation of our equipment.
- The quality of figure 4 (A-C) must be increased because the colocalization is not well distinguished
Ans: Thank you for the reviewer’s comment. We have removed Figure 4(A-C) in the revised manuscript. In addition, we have merged the Figure 4 and Figure 5 into one figure in the revised manuscript, which describes the role of CD14 in the phagocytosis of apoptotic cells more clearly.
Reviewer 2 Report
In the present study, Lin et al. demonstrated that apoptotic ATRA-NB4 cells release CD14+ apo-MPs which trigger the phagocytic activity of viable ATRA-NB4 cells, thereby contributing to the engulfment of the apoptotic cells. Although the study is novel and interesting, there are a lot of concerns which need to be taken care of. The comments are shown below:
1. In the present study the authors showed the effect of microparticles (MPs) only, released from apoptotic CD14+ neutrophils. Why the authors exclude the effect of other extracellular vesicles class, exosomes in this process. Numerous studies show the important contribution of exosomes in various pathophysiological conditions.
2. The introduction should be included with a brief description of extracellular vesicles.
3. Materials and Methods: page 3, 2.3. MP preparation and Flow Cytometric Analysis: line 129: ‘….sizes varying approximately between 0.1 and 2m….’. Is this a typo? Please correct.
4. Results: page 4, line 164: Figure 1 should be Figure 1A; line 169: Figure 1 should be Figure 1B.
5. Figure 1A and 1B: Why the counts of CD14 and TLR4 are decreased upon ATRA treatment as compared to control? Did the authors normalize the events or volume in these flow cytometric experiments?
6. Page 5, line 184: Figure 2 should be Figure 2A-C.
7. Cytochalasin D is well-known for disrupting the actin filaments and actin polymerization. Is the reduced phagocytic activity with Cytochalasin D a result of actin cytoskeleton disorganization?
8. How NF-κB and IRF3 are involved in the process? The authors should show the activation or induced expression by western blotting. Otherwise, it is not clear how these signaling molecules act and how they are related to each other.
9. Figure 2A-C: From the images, it appears that not every cell took up the labelled beads. What is the percentage of phagocytic cells among all the cell population. What functions the non-phagocytic cells play in this context?
10. For multiple group analysis, the comparison of a single group with all the other groups should be carried out using ANOVA followed by Dunnett post hoc test, as in Figure 3 and others. Please rectify.
11. The authors should provide a schematic representation for better understanding the signaling event.
12. Figure 4A-C: The picture appeared hazy and only a single cell among all the cell population seems to be phagocytic from the merged image. The authors should discuss the percentage of phagocytic cells.
13. The authors should show the uptake of MPs by the recipient cells. May be the authors label the MPs with PKH26 followed by showing the labelled MPs fluorescence into the recipient cells along with DAPI to stain the nucleus.
14. Is the enhanced expression of CD14 in MPs-fused ATRA-NB4 viable cells a result of MPs transferred CD14 to the recipient cells or MPs-mediated induction of endogenous CD14 expression in the recipient cells? The best way to determine this is to perturb the protein synthesis in the recipient cells by cycloheximide followed by MPs incubation and check for CD14 level.
15. Does the pre-treatment of CD14 antibody block the incorporation of the MPs into the recipient cells leading to a reduction of phagocytic index in Figure 6D? Please explain.
In some places it was not very clear, hence additional editing of the English is required.
Author Response
In the present study, Lin et al. demonstrated that apoptotic ATRA-NB4 cells release CD14+ apo-MPs which trigger the phagocytic activity of viable ATRA-NB4 cells, thereby contributing to the engulfment of the apoptotic cells. Although the study is novel and interesting, there are a lot of concerns which need to be taken care of. The comments are shown below:
- In the present study the authors showed the effect of microparticles (MPs) only, released from apoptotic CD14+ neutrophils. Why the authors exclude the effect of other extracellular vesicles class, exosomes in this process. Numerous studies show the important contribution of exosomes in various pathophysiological conditions.
Answer: We fully agree with the reviewer’s opinion at the crucial role of exosomes in the cell-cell interaction in various pathophysiological condition. Based on our previous study (as shown below), it is likely that part of exosomes are also harvested in the microparticles concentrate in the current study. Further study by using exosomes-specific harvesting method is crucial in studying the role of CD14(+) exosomes in the phagocytosis of apoptotic cells.
WH Tsai et al. Annexin A1 mediates the anti-inflammatory effects during the granulocytic differentiation process in all-trans retinoic acid-treated acute promyelocytic leukemic cells. J Cell Physiol 2012 Nov;227(11):3661-9. doi: 10.1002/jcp.24073.
- The introduction should be included with a brief description of extracellular vesicles.
Ans: Thanks for your suggestion. We have added a paragraph to introduce the crucial role of microparticles in the cell-cell interaction during inflammation process, as shown in L69 - L77.
- Materials and Methods: page 3, 2.3. MP preparation and Flow Cytometric Analysis: line 129: ‘….sizes varying approximately between 0.1 and 2m….’. Is this a typo? Please correct.
Ans: We have corrected the typing error, as shown “ between 0.1 and 2 mm” in L151.
- Results: page 4, line 164: Figure 1 should be Figure 1A; line 169: Figure 1 should be Figure 1B.
Ans: We have corrected accordingly.
- Figure 1A and 1B: Why the counts of CD14 and TLR4 are decreased upon ATRA treatment as compared to control? Did the authors normalize the events or volume in these flow cytometric experiments?
Ans: We have reported previously that the proliferation of NB4 cells is suppressed after ATRA treatment in a time-dependent and ATRA-dosage dependent manner, which results in the decreased cell number of ATRA-NB4 cells, as compared with those NB4 cells without treatment. No volume normalization was applied in our study.
Hsu, H.C., et al., In vitro effect of granulocyte‐colony stimulating factor and all‐trans retinoic acid on the expression of inflammatory cytokines and adhesion molecules in acute promyelocytic leukemic cells. European journal of haematology, 1999. 63(1): p. 11-18.
- Page 5, line 184: Figure 2 should be Figure 2A-C.
Ans: We have revised accordingly.
- Cytochalasin D is well-known for disrupting the actin filaments and actin polymerization. Is the reduced phagocytic activity with Cytochalasin D a result of actin cytoskeleton disorganization?
Ans: We fully agree with your speculation, and this deserves further study to proved it.
- How NF-κB and IRF3 are involved in the process? The authors should show the activation or induced expression by western blotting. Otherwise, it is not clear how these signaling molecules act and how they are related to each other.
Ans: We fully agree with the reviewer’s opinion. Our current study has clearly demonstrated that NF-κB and IRF3 are involved in ATRA-NB4 cells while they are undertaking the phagocytosis of latex beads or apoptotic cells. This warrants further western blotting studies to investigate the differential effect of both signaling molecules in the induction of phagocytic activity of NB4 cells after ATRA treatment.
- Figure 2A-C: From the images, it appears that not every cell took up the labeled What is the percentage of phagocytic cells among all the cell population. What functions the non-phagocytic cells play in this context?
Ans: As shown in Figure 2D, phagocytic engulfment of latex beads were found in around 15% of ATRA-NB4 cells, this also implies that the phagocytic activity among neutrophilic precursor cells during the granulocytic differentiation process. Based on previous clinical studies, ATRA-induced granulocytic differentiation needs longer duration of ATRA treatment (> 7-14 days) to induce acute promyelocytic leukemic cells (such as NB4 cells) into mature granulocytes. Therefore, those non-phagocytic ATRA-NB4 cells are not mature enough to gain phagocytic activity.
Meng-Er, Huang, et al. (1988). "Use of all-trans retinoic acid in the treatment of acute promyelocytic leukemia." Blood 72(2): 567-572.
- For multiple group analysis, the comparison of a single group with all the other groups should be carried out using ANOVA followed by Dunnett post hoc test, as in Figure 3 and others. Please rectify.
Ans: Thank you very much for the reviewer’s comment. We have reviewed our raw data statistical analysis method in which Dunnett post hoc test was also used in part of analysis. We have added it in the revised manuscript as shown in L184.
- The authors should provide a schematic representation for better understanding the signaling event.
Ans: Thank you very much for the reviewer’s suggestion. Our current study does point out the involvement of NF-κB and IRF3 in the phagocytic activity of ATRA-NB4 cells. We wish we can make the signaling pathway schematic more informative in the near future when the Western blotting data are available.
- Figure 4A-C: The picture appeared hazy and only a single cell among all the cell population seems to be phagocytic from the merged image. The authors should discuss the percentage of phagocytic cells.
Ans: Thank you for your suggestion. We have demonstrate the percentage of cells with phagocytic activity in the “result” part of revised manuscript as shown in L328-L332 & L339-L343.
- The authors should show the uptake of MPs by the recipient cells. May be the authors label the MPs with PKH26 followed by showing the labelled MPs fluorescence into the recipient cells along with DAPI to stain the nucleus.
Ans: Figure 5 in our revised manuscript has demonstrated that CD14 expression and phagocytic activity are both enhanced when viable ATRA-NB4 cells have pre-incubated with apo-MP. In addition, the apo-MP-enhanced phagocytic activity was significantly attenuated when those apo-MP were pretreated with anti-CD14 antibody before incubation with viable ATRA-NB4 cells for phagocytic assay. These evidences strongly indicate that CD14(+)apo-MP, derived from apoptotic ATRA-NB4 cells, are responsible for the increase of surface CD14 expression (Figure 5B&C) as well as the phagocytic activity(Figure 5D) in the recipient viable ATRA-NB4 cells. Based on these evidences, the reviewer’s concern is least likely.
- Is the enhanced expression of CD14 in MPs-fused ATRA-NB4 viable cells a result of MPs transferred CD14 to the recipient cells or MPs-mediated induction of endogenous CD14 expression in the recipient cells? The best way to determine this is to perturb the protein synthesis in the recipient cells by cycloheximide followed by MPs incubation and check for CD14 level.
Ans: We fully agree with the reviewer’s opinion and the suggestion in designing further studies in investigating the interaction between CD14(+)apo-MP and viable ATRA-NB4 cells. The fusing/adhesion mechanism is proposed by our group because we have demonstrated that both CD14(+)apo-MP and CX3CL1(+)apo-MP are responsible for the transfer of CD14 and CX3CL1 from apoptotic ATRA-NB4 cells to the viable ones. However, we still need further studies to prove our proposal.
Tsai, W.-H., et al., CX3CL1 (+) Microparticles-Induced MFG-E8 Enhances Apoptotic Cell Clearance by Alveolar Macrophages. Cells, 2021. 10(10): p. 2583.
Tsai, W.-H., et al., CX3CL1(+) Microparticles Mediate the Chemoattraction of Alveolar Macrophages toward Apoptotic Acute Promyelocytic Leukemic Cells.Cell Physiol Biochem 2014;33:594-604
- Does the pre-treatment of CD14 antibody block the incorporation of the MPs into the recipient cells leading to a reduction of phagocytic index in Figure 6D? Please explain.
Ans: In the current and previous studies, we have demonstrated that apo-MP derived from apoptotic ATRA-NB4 cells play a crucial role in the transfer of both CD14 and CX3CL1 from apoptotic cells to the viable ATRA-NB4 cells, resulting in the increased surface expression of both CD14 and CX3CL1 in the recipient cells. We still need further studies to investigate the mechanism underlying the uptake of apo-MP by the viable ATRA-NB4 cells. The reviewer’s hypothesis is quite interesting, which also warrant investigation.
Comments on the Quality of English Language
In some places it was not very clear, hence additional editing of the English is required.
Ans: We thanks for the reviewer’s comment. The other reviewer have pointed out the typing errors in our English, and we have corrected them accordingly. We have also sent our revised manuscript for English editing,
Round 2
Reviewer 2 Report
The authors adequately addressed all of my concerns.
English has been improved.